Effects of wheat intercropping on growth and occurrence of Fusarium wilt in watermelon

Lv HuiFang 1 2
Yan CongSheng 1 congshengyan@126.com
1 Institute of Vegetables, Anhui Academy of Agricultural Sciences , Heifei, Anhui , China
2 Blueberry Engineering Technology Research Center of Anhui, School of Biology and Food Engineering, HeFei Normal University , Hefei, Anhui , China
Cao Yunpeng
Electronic publication date: 2024 Jun 28
Publication date: 2024
Volume: 12
Electronic Location ID: e17587
Received 2023 Dec 21; Accepted 2024 May 28
Copyright: © 2024 Lv and Yan
Copyright year: 2024
Copyright holder: Lv and Yan
License: This is an open access article distributed under the terms of the Creative Commons Attribution License, which permits unrestricted use, distribution, reproduction and adaptation in any medium and for any purpose provided that it is properly attributed. For attribution, the original author(s), title, publication source (PeerJ) and either DOI or URL of the article must be cited.
License URL: https://creativecommons.org/licenses/by/4.0/

Keywords: Wheat, Watermelon, Fusarium wilt, Photosynthesis, Antioxidant enzyme, Soil enzyme, Soil microbes

Funding: China Agriculture Research System CARS-25 Anhui Province Key Research and Development Program 2023j11020002 University Natural Science Research Project of Anhui Province 2023AH051284 HeFei Normal University 2022rcjj02 Anhui Academy of Agricultural Sciences QNYC-201906 This study was supported by the China Agriculture Research System (CARS-25), Anhui Province Key Research and Development Program (2023j11020002), the University Natural Science Research Project of Anhui Province (2023AH051284), the Scientific Research Startup Fund for High-level Talents, HeFei Normal University (2022rcjj02), and the Yong Talents Fund, Anhui Academy of Agricultural Sciences (QNYC-201906). The funders had no role in study design, data collection and analysis, decision to publish, or preparation of the manuscript.

==============================
Watermelon is commonly affected by Fusarium wilt in a monoculture cropping system. Wheat intercropping alleviates the affection of Fusarium wilt of watermelon. The objective of this study was to determine the effects of wheat and watermelon intercropping on watermelon growth and Fusarium wilt. Our results showed that wheat and watermelon intercropping promoted growth, increased chlorophyll content, and photosynthesis of watermelon. Meanwhile, wheat and watermelon intercropping inhibited watermelon Fusarium wilt occurrence, decreased spore numbers, increased root vigor, increased antioxidant enzyme activities, and decreased malondialdehyde (MDA) content in watermelon roots. Additionally, wheat and watermelon intercropping enhanced the bacterial colonies and total microbes growth in soil, decreased fungi and Fusarium oxysporum f. sp. niveum (FON) colonies, and increased soil enzyme activities in watermelon rhizosphere soil. Our results indicated that wheat and watermelon intercropping enhanced watermelon growth and decreased the incidence of Fusarium wilt in watermelon. These effects could be due to intercropping inducing physiological changes, regulating soil enzyme activities, and/or modulating soil microbial communities.

Introduction

Watermelon (Citrullus lanatus L.) is one of the most popular crops cultivated worldwide. China is the world’s largest producer of watermelon (He et al., 2022). However, the continuous cropping of watermelon is becoming increasingly common, which often results in disease pressure becoming more intense. For instance, watermelon wilt disease is caused by the soil-borne pathogen Fusarium oxysporum F. sp. Niveum (FON), which leads to the plant death and reduced watermelon yield (Ren et al., 2008). In agroecosystems, intercropping is beneficial for plant growth (Xiao et al., 2013), increases production (Kermah et al., 2017; Wu et al., 2022; Zhu et al., 2022), and suppressess some soil-borne diseases, such as Fusaria, Cylindrocladium parasiticum, Phytophthora capsici, Verticillium dahliae (Gao et al., 2014; Yang et al., 2014; Xu, Wang & Wu, 2015a; Li et al., 2019; Zhou et al., 2023). Nevertheless, the intricate process through which intercropping inhibits soil-borne infections involves root exudates (Fu et al., 2015; Li et al., 2019), enzymes (Li et al., 1999; Lv et al., 2023), soil microorganisms (Zhou et al., 2023), and pathogen density (Ren et al., 2008; Xu, Wang & Wu, 2015a).

Photosynthesis is the basis of crop growth and yield. As a photosynthetic pigment, chlorophyll is essential for the absorption of light energy by plants, and the content of chlorophyll directly affects the intensity of plant photosynthesis (Kumagai, Araki & Kubota, 2009; Zhu et al., 2016). Photosynthetic rate, intercellular CO2 concentration, stomatal conductance, and transpiration rate are important indices of photosynthesis. Several studies have shown that intercropping improved the utilization of light energy and increased the yield per unit area in different crops (Ai & Ma, 2020; Li et al., 2014, 2020; Wang et al., 2021).

Malondialdehyde (MDA) is one of the final products of membrane lipid peroxidation in plants and a vital indicator of oxidative stress (Sharma et al., 2012). Superoxide dismutase (SOD) is one of the antioxidant enzymes that scavenge the superoxide anion by converting this free radical to O2 and H2O2. Catalase (CAT) is a tetrameric antioxidant enzyme that catalyzes the conversion of H2O2 to H2O and O2 (Yin et al., 2019). Ascorbate peroxidase (APX) is an enzyme that converts the H2O2 into H2O using ascorbate as a substrate to decrease its potency (Noctor & Foyer, 1998). When plants are subjected to stress, the antioxidant enzymes, such as CAT, SOD, and APX, play essential roles in maintaining the dynamic balance of free radical production and removal, thus preventing reactive oxygen accumulation and further damage. The antioxidant enzymes like SOD, CAT, and APX are essential enzymes for scavenging reactive oxygen species in plants, which has a positively correlated with resistance to disease (Mandal, Mitra & Mallick, 2008; Lv et al., 2023).

Plant disease resistance may be influenced by local biotic environmental factors, such as plant-associated microbiota (Choi et al., 2020; Vannier et al., 2019). Rhizosphere microorganisms, considered the second genome of plant, play a crucial role in modulating the host immune system (Zamioudis & Pieterse, 2012), altering the defense capabilities of plants (Zhang et al., 2011; Lanoue et al., 2010) or adding plant beneficial bacteria (Yang et al., 2011; Lee, Lee & Ryu, 2012). Previous studies showed that the rhizosphere microorganisms play a vital role in plant disease resistance (Berendsen et al., 2018; Kwak et al., 2018). Rhizosphere microbiome significantly contributes to plant disease resistance (Li et al., 2019; Wei et al., 2015), and decreases the pathogen abundance.

Soil enzymes, primarily derived from plant roots, microorganisms, and animal residues, indicate soil’s potential to facilitate biochemical processes like organic residue decomposition and nutrient cycling (Lalande et al., 2000; Casucci, Okeke & Frankenberger, 2003). Studies have shown that soil enzyme activities can serve as a reliable indicator of soil quality (Gianfreda et al., 2005; Acosta-Martínez et al., 2007).

Soil sucrase, urease and phosphatase activities were increased for eggplant with green garlic intercropping compared to monoculture during the different intercropping years (Wang et al., 2014), and the legume-intercropped maize showed higher activities than those grown under the monocropping system (Zhang et al., 2013). A previous study reported that the soil alkaline phosphatase activity in the rhizosphere soil of tomatoes was higher under garlic-tomato intercropping systems than those grown in monoculture (Liu et al., 2014). Likewise, intercropping with aromatic plants increased organic and available nitrogen in the soil and increased the soil protease and urease activities in an orchard ecosystem (Chen et al., 2014).

As a common diversified planting system, intercropping systems, particularly allelopathic crops, have been widely used in the sustainable production of farmland. In particular, the use of allelopathic crops, such as wheat and rice, are widely used for sustainable farmland production and are known to suppress soil-borne diseases (Ren et al., 2008; Xu, Wang & Wu, 2015a). Fusarium wilt is one of the most common diseases in watermelon production, resulting in poor growth and reduced watermelon yield and quality. Wheat is one of the most important cereal crops all over the world. Wheat roots secrete various allelopathic compounds, such as coumaric acid, which inhibits the growth of the pathogen of watermelon Fusarium wilt (Lv et al., 2018). Wheat intercropping with watermelon is used to promote disease suppression in watermelon plants and is extensively employed as a proactive strategy to control Fusarium wilt in watermelon. Past studies have shown that the growth of FON was significantly inhibited by wheat root exudates (Xu et al., 2015b; Lv et al., 2018). However, in the wheat-watermelon intercropping system, it is unclear whether wheat root exudates can be transferred to the rhizosphere of watermelon and its impact on the growth and occurrence of Fusarium wilt of watermelon.

Herein, we evaluated the impact of watermelon and wheat intercropping with different root separations on the Fusarium wilt of watermelon. We measured leaf photosynthetic parameters and assessed the effects of watermelon and wheat intercropping on enzymatic activities associated with plant disease resistance. Moreover, we determined the changes in soil microbial amount and enzymatic activities in the rhizosphere soil of watermelon.

Materials and Methods

Plant materials and pathogen isolates

The current experiment was performed in a greenhouse at the National Center for Vegetable Improvement, Huazhong Agricultural University, China. Zaojia 8424 and Emai 18 seeds were preserved and used from our previous studies (Lv et al., 2023, 2018). Fusarium oxysporum f. sp niveum (FON) (race-1) was isolated from an infected watermelon plant provided by Zhengzhou Fruit Research Institute, Chinese Academy of Agricultural Sciences.

Experimental treatments

In the current study, three treatments were included: (1) watermelon monocropping only one watermelon plant per pot as a control (CK), (2) watermelon and wheat intercropping (NS) with non-separation roots (watermelon and wheat were planted in a pot, and the roots have a common rhizosphere by full interaction), and (3) watermelon and wheat intercropping (MS) with root nylon mesh separation (watermelon and wheat were planted in a pot, and the two roots were separated by a nylon mesh with 30 μm aperture, and the roots cannot pass through nylon mesh, but root exudate, nutrients, and water could pass through).

After being submerged in water for 5 h, watermelon seeds were incubated for 2 days at 30 °C in a growth chamber. After that, the sprouted seeds were planted in a plug plastic tray containing a mixture of peat and vermiculite (v/v, 1:1). To simulate continuous cropping, 8 ml conidial suspension of FON (106 mL−1, 100 mL per pot) were inoculated into each pot substrate to enhance the pathogenicity of the substrate before watermelon transplanting. Seedlings at the four-leaf stage were transferred into plastic pots (34 cm diameter, 24 cm height) filled with 8 L of a mixed substrate. Watermelon and wheat were grown on either side of the pot. Forty wheat seedlings were grown alongside one watermelon seedling, with a 10 cm gap between them. In the experiment, a total of 90 pots containing three replicates of each treatment were randomly let out. The plants’ nutritional needs were met by using the Hoagland solution (Hoagland & Arnon, 1950).

Chlorophyll content, root vigor, plant biomass, and leaf photosynthetic parameters were measured at three stages: 15, 25, and 35 days after transplanting. Watermelon root samples were immediately collected in liquid nitrogen and were stored at −80 °C for further analysis. Root samples were utilized to analyze the antioxidant enzyme activity and MDA content. The rhizosphere soil samples were collected at 15, 25, and 35 days after transplanting and then stored at −80 °C for the quantification of the FON population, soil enzyme activity, and microbial amount.

Determination of chlorophyll content and root vigor

Chlorophyll content was measured by 96% ethanol extraction colorimetric according to the previously described method (Liu et al., 2020). Root vigor was determined with triphenyltetrazolium chloride per the procedure described by Zhong et al. (2019).

Determination of antioxidant enzyme activity and MDA content

Root tissues (0.3 g) were ground in 2 mL ice-cold 50 mmol·L−1phosphate buffer (PBS), with pH 7.8, that contained 0.2 mmol·L−1 EDTA and 2% (w/v) polyvinylpyrrolidone. The obtained homogenates were centrifuged at 12,000 × g for 20 min, and then the resulting supernatants were stored at −4 °C for further analysis of antioxidant enzyme activity and MDA content.

MDA content was evaluated with thiobarbituric acid following a previously described method (Hodges et al., 1999). CAT activity was measured by the ammonium molybdate spectrophotometric method (Kosar et al., 2021), and SOD activity was evaluated as described in a previous report by García-Triana et al. (2010). APX activity was determined by examining the decline in A290 nm (Nakano & Asada, 1981).

Determination of leaf photosynthetic parameters

The net photosynthetic rate (Pn), intercellular CO2 concentration (Ci), transpiration rate (Tr), and stomatal conductance (Gs) were measured with an open gas-exchange system (LI-COR 6800; Lincoln, Dearborn, MI, USA) using the 14th, 16th, and 20th leaf from each plant at 15, 25, and 35 days after transplanting.

Determination of plant biomass and single fruit weight of watermelon

At 15, 25, and 35 days after transplanting, six plants were harvested per replicate of each treatment, roots were washed with water, and then the fresh weight of the plants was measured by electric balance. The cleaned plants were retained in an oven at 80 °C for 3 days before being weighed. The single fruit weight of the watermelon was harvested and then measured by electric balance 35 days after transplanting.

Determination of incidence of wilt disease

The incidence of watermelon wilt disease was measured 15 days after transplanting and recorded after every 10 days, as described in a previously published method (Wu et al., 2009).

DNA extraction and quantitative real-time polymerase chain reaction

The DNA was extracted from the rhizosphere soils of watermelon in monocropping and intercropping groups using a PowerSoil DNA isolation kit (Omega Bio-Tek, Inc., Norcross, GA, USA). The specific primers FON-1 (5′-CGATTAGCGAAGACATTCACAAGACT-3′) and FON-2 (5’-ACGGTCAAGAAGATGCAGGGTAAAGGT-3′) were designed according to the described method (Zhang & Wang, 2005). The reverse transcription quantitative real-time PCR (qRT-PCR) was performed on QuantStudio (TM) seven Flex Real-Time PCR System with polymerase chain reaction (PCR) amplification (Lv et al., 2018). The standard curves were calculated as described by Zhou & Wu (2012). The number of copies was used to calculate the FON population (Wakelin et al., 2008).

Soil microbe analysis

The culturable fungi, bacteria, and actinomycetes in rhizosphere soil were counted using the dilution plate method as described previously by Cheng et al. (2000). In brief, 1.0 g of soil sample was put in a triangular flask containing 9 ml sterile water and shake for 20 min. 1 ml of soil suspension was added into a test tube containing 9 ml sterile water, then make 10−2, 10−3, 10−4, 10−5, 10−6, 10−7 different dilution gradients of the soil solution. 100 µl of the diluted soil suspension was spread on Petri dishes with agar medium and incubated to obtain single colonies. The number of Fusarium oxysporum was determined using the pentachloronitrobenzene (PCNB) medium with dilution gradients of 10−2, 10−3, 10−4 at 28 °C for 5 days. The number of fungi was determined using Martin’s agar medium with dilution gradients of 10−2, 10−3, 10−4 at 28 °C for 5 days. Bacterial populations were assayed using the beef extract peptone medium with dilution gradients of 10−5, 10−6, 10−7 at 30 °C for 2 days. The number of actinomycetes was determined using the Gause 1 medium with dilution gradients of 10−3, 10−4, 10−5 at 30 °C for 4 days.

Determination of soil enzyme activities

This study determined the activities of soil sucrase, proteinase, urease, acid phosphatase, peroxidase, and dehydrogenase (Guan, 1986). Soil sucrase activity was determined by incubating 1 g of soil with the formation of glucose for 24 h at 37 °C. The supernatants reacted with 3,5-dinitrosalicylic acid were detected at a wavelength of 508 nm. Soil protease activity was determined by ninhydrin colorimetry. The released glycine was measured at a wavelength of 560 nm. Soil urease activity was determined by sodium phenol-sodium hypochlorite colorimetry. The formation of NH3-N was determined at the wavelength of 578 nm. Acid phosphatase activity was determined by disodium p-nitrobenzene phosphate and released p-nitrophenol was measured at a wavelength of 410 nm. Dehydrogenase was determined by incubating 1 g of soil with 2,3,5-chlorotriphenyltetrazolium chloride as a substrate for 6 h at 37 °C. The formation of TPF was measured at 485 nm. Peroxidase activity was determined by pyrogallol colorimetry by incubating 1g soil with the formation of epigallocatechin for 24 h at 37 °C and was determined at 430 nm.

Statistical analysis

The SPSS (IBM 19, Armonk, NY, USA) was used to analyze the statistical significance of the data, and Tukey’s test was used to compute the differences between treatment means at the 5% level. All the values were presented as mean ± standard error.

Results

Effect of wheat and watermelon intercropping with different root separation treatments on chlorophyll content and root vigor of watermelon

To understand the effect of wheat-watermelon intercropping on the chlorophyll content and root vigor of watermelon, we measured the chlorophyll contents and root vigor at different stages of treatment in watermelon. Our results revealed that watermelon leaf chlorophyll content and root vigor in NS and MS treatments were significantly higher compared with their control (CK) at 15, 25, and 35 days after transplanting (Fig. 1). In particular, the root vigor was increased by up to 1.7-fold and 1.9-fold in NS and MS treatments compared with the CK at 35 days after transplanting (Fig. 1B).

Figure 1 Changes of chlorophy II content (A) and root vigor (B) in watermelon during different treatments.

The values were mean ± SE of three replicates for each treatment. The different letters “a, b, c” of the same day represented the significant differences at P < 0.05 by Tukey’s test.

Effect of wheat and watermelon intercropping with different root separation treatments on leaf photosynthesis parameters of watermelon

Photosynthesis is a well-studied process in which plants convert light energy into chemical energy and transform carbon dioxide to produce organic molecules, which are then used for plant growth and development. Herein we analyzed the photosynthesis parameters of the watermelon leaf. The leaf photosynthetic rate in NS and MS treatments were significantly higher compared with the CK at 15, 25, and 35 days after transplanting (Fig. 2A). The intercellular CO2 concentration and stomatal conductance of watermelon leaf in NS treatment were increased from 1.2 to 1.3 and 1.3 to 1.9 fold compared with the CK at 15 to 35 days after transplanting. Similarly, the intercellular CO2 concentration of watermelon leaf in MS treatment was significantly higher than that of CK at 15 and 25 days after transplanting by 20.7% and 22.4%, respectively. The stomatal conductance of the watermelon leaf also showed a similar trend as the photosynthesis rate and CO2 concentration, it was significantly higher in NS and MS treatments at 15, 25, and 35 days in comparison with the CK treatment (Figs. 2B and 2D). Nonetheless, there were no appreciable variations in the watermelon leaf’s transpiration rate across the various treatments (Fig. 3C).

Figure 2 Changes of leaf photosynthesis parameters of watermelon in different treatments.

(A) The photosynthetic rate. (B) The intercellular CO2 concentration. (C) The transpiration rate. (D) The stomatal conductance. The values were mean ± SE of three replicates for each treatment. The different letters “a, b, c” of the same day represented the significant differences at P < 0.05 by Tukey’s test.

Figure 3 Changes of CAT activity (A), SOD activity (B), APX activity (C), and MDA content (D) of watermelon in different treatments.

The values were mean ± SE of three replicates for each treatment. The different letters “a, b, c” of the same day represented the significant differences at P < 0.05 by Tukey’s test.

Effect of wheat and watermelon intercropping with different root separation treatments on antioxidant enzyme activities and MDA content of watermelon

After the watermelon plants intercropped with wheat plants at 15, 25, and 35 days, the CAT and APX activity levels in NS and MS treatments were statistically higher relative to the CK treatment (Figs. 3A and 3C). The SOD activity was significantly increased at 15, 25, and 35 days after transplanting in the NS treatment relative to the CK treatment. Likewise, the SOD activity was significantly increased at 15 and 35 days after transplanting in the MS treatment, but at 25 days after transplanting, we observed no significant differences in SOD activity compared to CK (Fig. 3B). APX activity was also significantly higher in NS and MS treatments compared to CK. Interestingly, at 15 days, the APX activity was higher in all three treatments compared to 25 and 35 days. However, MDA content in the NS and MS treatments was significantly lower in comparison with the CK treatment (Fig. 3D).

Effect of wheat and watermelon intercropping with different root separation treatments on growth and yield of watermelon

The photosynthetic rate and enzymatic activities directly influence the growth and yield of crops. To know the effect of wheat-watermelon intercropping on watermelon growth and yield. We weighed the watermelon in different treatments. At 15 days after transplanting, the dry and fresh weight of the watermelon showed no significant differences among the NS, MS, and CK treatments (Figs. 4A and 4B). However, 35 days after transplanting, we observed that the dry weight (31.2 plant.g −1, 32.0 plant.g −1) and fresh weight (300.7 plant.g −1, 301.6 plant.g −1) obtained from the NS and MS treatment was significantly higher compared with the CK treatment (265.7 plant.g −1, 27.6 plant.g −1) (Figs. 4D and 4E). Similarly, At 35 days after transplanting, the single fruit weight of watermelon in the NS and MS treatment was significantly greater relative to the CK treatment (CK) (Figs. 4C and 4F).

Figure 4 Changes of growth and fruit weight of watermelon in different treatments.

(A, B) Different treatments of watermelon. (C) The phenotypes of watermelon at 35 day after transplanting. (D) The fresh weight of watermelon plants. (E) The dry weight of watermelon plants. (F) The single fruit weight of watermelon. The values were mean ± SE of three replicates for each treatment. The different letters “a, b, c” of the same day represented the significant differences at P < 0.05 by Tukey’s test.

Effect of wheat and watermelon intercropping with different root separation treatments on the incidence rate of Fusarium wilt in watermelon

The dilution plate counting method and qRT-PCR technique were used to quantify the amount of FON in the watermelon rhizosphere. At three stages, 15, 25, and 35 days after transplanting, watermelon plants raised in soil with wheat and watermelon intercropping (NS and MS) exhibited a significantly lower number of FON in comparison with watermelon plants raised in soil with monocropping (CK) (Figs. 5A and 5B). Watermelon plants grown in soil with wheat-watermelon intercropping with root non-separation (NS) showed no significant differences compared with wheat and watermelon intercropping with root mesh separation soil (MS).

Figure 5 Occurrence of Fusarium wilt of watermelon different treatments.

(A and B) the number of FON in the rhizosphere of watermelon. (C) Occurrence of Fusarium wilt of watermelon. The values were mean ± SE of three replicates for each treatment. The different letters “a, b, c” of the same day represented the significant differences at P < 0.05 by Tukey’s test.

The analysis of Fusarium wilt in watermelon revealed that the incidence rate of Fusarium wilt in watermelon differed from the effects of intercropping wheat- watermelon with distinct root separation treatments (Fig. 5C). The incidence of watermelon trend was similar to that for the number of FON among all treatments. At 15 days after transplanting, watermelons in the CK treatment began to show Fusarium wilt symptoms. Similarly, several watermelon plants exhibited symptoms of Fusarium wilt in the MS treatment 19 days after transplanting. However, the watermelon plants grew normally in the NS treatment. At 35 days after transplanting, the incidence rate of Fusarium wilt in NS and MS treatments was significantly lower (43.3% and 45.0%) compared with the CK treatment (64.4 %) (Fig. 5C). Our results revealed that the wheat-watermelon intercropping reduced the occurrence rate of Fusarium wilt on watermelon.

Effect of wheat and watermelon intercropping with different root separation treatments on microbial communities in rhizosphere soil of watermelon

Plant rhizosphere microbial communities play a crucial role in the conversion of soil organic matter, which in turn influences plant growth. Here, we investigated the microbial communities in the watermelon plant rhizosphere. The results revealed that at 15, 25, and 35 days after transplanting, the number of soil fungi in the watermelon rhizosphere was significantly higher in the CK treatment compared with the NS and MS treatments. At 15 days after transplanting, the number of fungi in the rhizosphere soil of watermelon in the CK treatment was 27 × 103 g−1 soil, while we observed only 12.7 × 103 and 12 × 103 g−1 soil in rhizosphere of plant with NS and MS treatments, respectively (Fig. 6A). The highest number of actinomycetes in the rhizosphere soil of watermelon was obtained in CK treatment at 15 days, followed by MS treatment at 25 days. At 35 days after transplanting, no significant differences were found among the NS, MS, and CK treatments (Fig. 6C). The number of bacteria in the rhizosphere soil of watermelon was significantly higher in the NS and MS treatments compared with the CK treatment at 15, 25 and 35 days (Fig. 6B). The tendency for the number of total microbes in the rhizosphere soil of watermelon was similar to that for the number of bacteria. The total amounts of microbes in the rhizosphere soil of watermelon in the NS and MS treatments were higher compared with the CK treatments at 15, 25, and 35 days after transplanting (Fig. 6D). Our results indicated that the wheat-watermelon intercropping enhanced the microbial communities in plant rhizosphere of watermelon.

Figure 6 Changes of microbial number in the rhizosphere soil of watermelon in different treatments.

(A) The number of fungi in rhizosphere soil of watermelon. (B) The number of bacteria in rhizosphere soil of watermelon. (C) The number of actinomycetes in rhizosphere soil of watermelon. (D) The number of total microorganism in rhizosphere soil of watermelon. The values were mean ± SE of three replicates for each treatment. The different letters “a, b, c” of the same day represented the significant differences at P < 0.05 by Tukey’s test.

Effect of wheat and watermelon intercropping with different root separation treatments on soil enzyme activity in watermelon plant rhizosphere

In wheat-watermelon intercropping with root separation treatment, our assesments revealed that, the initial periods 15 and 25 days after transplanting, the soil sucrase content in the plant rhizosphere of watermelon was increased in the NS and MS treatments compared with the CK treatment, but decreased lately at 35 days after transplanting (Fig. 7A). The soil protease in the rhizosphere soil of watermelon obtained with the NS and MS treatments were elevated compared to the CK treatment at 15 days after transplanting. However, no significant differences were found among the three treatments at 25 days after transplanting. At 35 days after transplanting, the soil protease in CK treatment was higher compared to the other treatments (Fig. 7B).

Figure 7 Changes of sucrase activity (A), proteinase activity (B), urease activity (C), phosphatase activity (D), peroxidase activity (E), dehydrogenase activity (F) in the rhizosphere soil of watermelon in different treatments.

The values were mean ± SE of three replicates for each treatment. The different letters “a, b, c” of the same day represented the significant differences at P < 0.05 by Tukey’s test.

At 15, 25, and 35 days after transplanting, the soil urease and peroxidase activities obtained from the plant’s rizosphere with the CK treatment were significantly decreased compared with the NS and MS treatments (Figs. 7C and 7E). At 15 days after transplanting, the soil phosphatase activity obtained with the CK treatment was lower than the NS and MS treatments. At 25 days after transplanting, the soil phosphatase activity showed no significant differences among the three treatments. Noteworthy, at 35 days after transplanting, the soil phosphatase activity also increased in the MS treatment (Fig. 7D). Furthermore, at 15 days after transplanting, no significant differences were observed in the soil dehydrogenase activity among the three treatments. In contrast at 25 days after transplanting, the soil dehydrogenase activities obtained with the NS and MS treatments were significantly increased compared with the CK treatments. However, the soil dehydrogenase activity was significantly higher in the CK treatment compared with the NS and MS treatments at 35 days after transplanting (Fig. 7F).

Discussion

A traditional planting pattern in agriculture, intercropping can help to realize the complementary and promoting relationship between crops by increasing the diversity of plant communities in farmland ecosystems. In the intercropping system, crops have different ecological niches with spatial reasonable relationships, so intercropping can effectively improve ecological resources such as light, water, fertilizer, gas, and heat (Echarte et al., 2011; Duchene, Vian & Celette, 2017) and thus is likely to improve crop yield. Previous studies demonstrate that reasonable intercropping is beneficial for plant growth and yield (Tilman, 2020; Yang et al., 2022). The intercropping cucumber with wheat or hairy vetch increased cucumber yield and improved soil health (Wang, Wu & Zhou, 2010). In comparison to monoculture maize, maize-soybean and maize-peanut intercropping had significantly higher specific leaf weight and chlorophyll contents (Fu et al., 2023). The dry matter achieved through the millet/cowpea intercropping was significantly higher than that achieved by either millet or cowpea mono-crop (Toudou, Daouda & Atta, 2023). The photosynthetic characteristics of plants in the intercropping system are usually as an essential indicator of plant growth. The photosynthetic rate was significantly higher in the maize with wheat intercropping than in sole maize (Gou et al., 2018). The current study, showed that the chlorophyll content, net photosynthetic rate, intercellular CO2 concentration, and stomatal conductance of watermelon leaf in NS and MS treatments were significantly increased compared with the CK treatment, and we observed an increase in fresh weight, dry weight and crop yield of watermelon plants at late growth period. A similar observation was also reported previously by Xu et al. (2014). The results of the current study indicated that the net photosynthetic rate of watermelon leaves was improved by increasing chlorophyll contents and by increasing intercellular CO2 concentration and stomatal conductance in the wheat and watermelon intercropping system, which proved to be beneficial for the growth and yield of the watermelon. Plants improved stress resistance by increasing root vigor (Huang et al., 2022). Intercropping tomato and garlic increased tomato root vigor (Li et al., 2012). Our results also revealed that the root vigor of watermelon in NS and MS treatments was significantly higher compared to their control (CK), probably because of allelochemical from wheat root exudates, which enhanced the resistance of watermelon to FON invasion.

MDA in plants increases biological membrane injury under stress conditions. Our study showed that MDA content was decreased in the NS and MS treatments compared with CK treatment. This result was consistent with a previous study (Ren et al., 2008). It is inferred that the wheat and watermelon intercropping exhibited solidity to biological membranes to prevent the watermelon from pathogenic infection by decreasing MDA content. CAT, APX, and SOD are evidenced to be essential antioxidant enzymes for scavenging reactive oxygen species, and their activities reflect the ability of cells to scavenge them (Stepien & Klobus, 2005). Several studies have revealed that the CAT, SOD, and APX activities in plants were increased to varying degrees under biotic stresses (Zhang et al., 2016; Chen et al., 2020). In the current study, the CAT, SOD, and APX activities were increased in the NS and MS treatments at 15, 25, and 35 days after transplanting. From these results, we assumed that the wheat root exudates may play a major role in enhancing watermelon resistance by improving antioxidant enzyme activities in wheat-watermelon intercropping systems, which might be a crucial reason for the reduced in the quantity of FON in the plant rhizosphere of watermelon and watermelon Fusarium wilt.

Crop cultivation systems, such as intercropping, rotation, and companion cropping, play important roles in the aboveground crop diversity and the diversity of rhizosphere soil microorganisms and provide resistance to biotic stresses. Soil microbes, such as bacteria, fungi, and actinomyces, are an important soil component. Soil microbes are considered the biological index of soil quality change. They involved in multiple ecological processes and significantly impact soil quality and function (Lagomarsin et al., 2011). Soil microbial communities high diversity, abundance, and species uniformity, are known to be associated with enhancing the resistance to soil-borne diseases (Bertin, Yang & Weston, 2003; Janvier et al., 2007; Bonilla et al., 2012). Brussaard, Ruiter & Brown (2007) reported that plant diseases are associated with microorganisms caused by some species of bacteria and actinomycetes, but fungi cause more damage. Boulter, Trevors & Boland (2002) also demonstrated that many bacteria found pathogen suppression. Dai et al. (2013) found that intercropping peanuts with Atractylodes lancea increased the bacterial community but inhibited the fungal community. Wang et al. (2015) found that the rotation of pineapple and banana reduced the number of pathogens in the soil and controlled the incidence of banana wilt disease effectively. The results of this study indicated that the number of soil bacterial communities and total microbes in plant the rhizosphere soil of watermelon were increased in the NS and MS treatments.

Interestingly, we observed that the number of soil fungi and FON in watermelon’s rhizosphere in the NS and MS treatments decreased. Disease incidence analysis also showed that the Fusarium wilt of watermelon was reduced in the NS and MS treatments, and these changes were related to wheat root exudations (Lv et al., 2018). From these results, we assumed that the environment formed by the microbes in the wheat-watermelon intercropping system was not suitable for FON growth and infection. In the wheat and watermelon intercropping systems, wheat root exudates act on the watermelon rhizosphere and play a rival role in delaying the occurrence of Fusarium wilt and reducing watermelon disease incidence by improving soil microbial activities. Our results were consistent with previous reports (Rudrappa et al., 2008; Ren et al., 2015; Berendsen et al., 2018; Carrión et al., 2019; Liu et al., 2021; Zhou et al., 2023), which demonstrated how plants adjust the chemistry of their root exudate to favor the recruitment of microorganisms that are beneficial to the plant in response to biotic stress (such as a pathogen or herbivore attacks).

Soil enzyme activity is an important component of soil quality. It can improve soil environment and nutrient cycling and promote the absorption of plant nutrients, which is closely related to plant health (Saha et al., 2008). Soil enzyme activities significantly altered in continuous monoculture systems, and the observed alterations include decreased soil urease and invertase activities in cucumber continuous monoculture (Wu, Meng & Wang, 2006). In the current study, we observed that at different watermelon growth stages, soil sucrase, protease, urease, dehydrogenase, peroxidase, and phosphatase activities in the NS and MS treatments were significantly increased compared to their control (CK). These findings demonstrated that intercropping systems have effects on soil enzyme activities at various growth stages. A previous study revealed that the soil sucrase activity was increased for peanuts with Atractylodes lancea intercropping compared to monoculture (Dai et al., 2013). The soil phosphatase activity in maize with peanut and soybeans intercropping was higher compared with the maize monocropping system (Yang et al., 2022). The improvement of soil enzyme activity is beneficial to the conversion and supply of soil nutrients during plant intercropping, and it can promote the growth and enzyme synthesis of soil microorganisms (Dick, 1992). Interestingly, in the current study, we found that wheat and watermelon intercropping contributes to increased soil enzyme activity in the plant rhizosphere of watermelon, involved in watermelon plant growth and suppressed the soil pathogen, which needs to be further studied.

Conclusions

This study indicated that wheat-watermelon intercropping significantly affects physiological changes and the soil characteristics of watermelon plants’ rhizosphere. The intercropping enhanced chlorophyll content, net photosynthetic rate, intercellular CO2 concentration, and stomatal conductance of watermelon leaf, root vigor, and the accumulation of biomass and single fruit weight. The antioxidants CAT, SOD, and APX activities of watermelon root were increased, while the MDA content was decreased in the wheat/watermelon intercropping system. The soil urease and peroxidase activities were increased, the number of soil bacteria and total microbes were increased, and the number of soil fungi and FON were decreased in the rhizosphere soil of the watermelon. Meanwhile, the amount of FON was reduced, which led to the reduction in Fusarium wilt occurance. Furthermore, as demonstrated by the encouragement of watermelon growth and the decrease in watermelon Fusarium wilt, the root separation treatment preserved the positive effects of intercropping. These results suggested that the increase in the growth and yield of the watermelon may be related to changes in chlorophyll content and photosynthesis, and the decrease in the incidence rate of Fusarium wilt may be related to changes in root vigor, antioxidant enzyme activities, MDA content, soil enzyme activities, microbial amount and FON relative abundance.

Supplemental Information

Supplemental Information 1 Raw data.

Supplemental Information 2 MIQE checklist.

Additional Information and Declarations

Competing Interests

Author Contributions

Data Availability

The authors declare that they have no competing interests.

HuiFang Lv conceived and designed the experiments, performed the experiments, analyzed the data, prepared figures and/or tables, authored or reviewed drafts of the article, and approved the final draft.

CongSheng Yan conceived and designed the experiments, authored or reviewed drafts of the article, and approved the final draft.

The following information was supplied regarding data availability:

The raw measurements are available in the Supplemental Files.

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
