# Peer review of "Effects of wheat intercropping on growth and occurrence of Fusarium wilt in watermelon"

_PeerJ, doi:10.7717/peerj.17587_

## Round 0.1 · original submission · Minor Revisions

Return to author for minor revisions.

**Language Note:** The review process has identified that the English language must be improved. PeerJ can provide language editing services - please contact us at [email protected] for pricing (be sure to provide your manuscript number and title). Alternatively, you should make your own arrangements to improve the language quality and provide details in your response letter. – PeerJ Staff

Reviewer 1 ·

Basic reporting

This manuscript written by Lv explores the effects of intercropping wheat and watermelon on the number and enzyme activity of microorganisms in the root system of watermelon. The results show that the intercropping system of wheat and watermelon can effectively improve watermelon yield and resistance to pathogens, which has practical application value. However, the English language in this manuscript is very poor. Extensive English editing is required. The literature used should be reviewed and some of the oldest items should be dropped and replaced with new items. In addition, the graphs are completely illegible - they need to be corrected so that the axis descriptions are clear.

Experimental design

The experimental design is relatively logical. However, the description of the materials and methods section requires the author to reorganize the language expression again.

Validity of the findings

Conclusions are well stated, linked to original research question & limited to supporting results.

Additional comments

I have listed only few of the corrections required as below.

Line 8: correct as “alleviate the affection of Fusarium wilt of”.
Line 8-11: This sentence is difficult to understand and needs to be revised.
Line 13: replace FON with spore.
Line 24: delete “and”.
Line 43-47: What is the purpose of this statement? This contradicts the previous claim that different crop combinations can increase yield per unit area.
Line54,56: Do CAT, SOD and APX belong to both ascorbase and antioxidant enzymes?
Line82: replace “while” with “and”.
Line85: replace “under” with “in”.
Line113: correct as ‘zaojia 8424’ and ‘Emai 18’.
Line128, 274: clarify what is meaning of “CFU” and “CPU”.
Line131: delete “from peat and vermiculite (v/v, 1:1)”.
Line147,155: use abbreviations directly, as they have already been mentioned.

Please carefully check the English spelling and grammar issues in the manuscript!

Reviewer 2 ·

Basic reporting

The manuscript by Lv et al. investigated the effects of intercropping wheat with watermelon to combat Fusarium wilt, a common issue in monoculture cultivation of watermelon. Through pot experiments, it was found that intercropping with wheat not only promotes watermelon growth by increasing chlorophyll content and photosynthesis rates but also significantly reduces the occurrence of Fusarium wilt. Key findings include a decrease in Fusarium oxysporum f. sp. niveum (FON) numbers, an increase in root vigor, enhanced activities of antioxidant enzymes, and a reduction in malondialdehyde (MDA) content in watermelon roots. Furthermore, this cultivation method beneficially altered soil microbial communities by increasing the number of bacteria and total microbes, reducing fungi and FON populations, and boosting soil enzyme activities in the rhizosphere of watermelon. The study concluded that wheat and watermelon intercropping effectively enhances watermelon growth and resilience against Fusarium wilt through physiological improvements, soil enzyme activity regulation, and microbial community modulation. Overall, the manuscript describes some important findings that are of interest to a broad audience.

Experimental design

1. Could you provide more details on the experimental setup for the intercropping system? Specifically, what was the spacing between wheat and watermelon plants, and how does this spacing compare to standard monoculture practices?
2. Were there any specific control measures in place to ensure that the observed effects on watermelon growth and Fusarium wilt reduction were solely due to the intercropping with wheat, and not other variables such as differences in soil nutrient content, watering regime, or pot size?
3. The study mentioned physiological changes, soil enzyme activities, and microbial community modulation as mechanisms through which wheat intercropping benefits watermelon. Could you elaborate on the specific physiological changes observed in watermelon plants and how these are directly linked to Fusarium wilt resistance?
4. Have there been any field trials to complement these pot experiments? It would be beneficial to understand if the positive effects of wheat and watermelon intercropping observed in a controlled environment hold true in larger-scale agricultural settings.
5. Could you provide more details on the techniques used to analyze soil microbial communities? Understanding the specific changes in microbial composition could offer insights into the biological control aspects of Fusarium wilt suppression.
6. From an agricultural economics perspective, how do the costs and labor associated with implementing a wheat and watermelon intercropping system compare to traditional monoculture cultivation and chemical control methods for Fusarium wilt?
7. Are there any insights or data on the long-term sustainability of using wheat and watermelon intercropping systems? Specifically, how might continuous intercropping affect soil health, watermelon yield, and disease resistance over multiple growing seasons?

Validity of the findings

1. How about the wheat plant growth in NS or MS settings? Are wheat plants all dead at 35 days after transplanting (Figure 4)?
2. For soil microbial community analysis, the important negative control (soil only, no plants) is missing.
3. For statistical analysis labeling, kindly consider employing the abc, a'b'c', a''b''c'' style to mitigate any potential confusion.

Additional comments

1. Line 1, Effect -> Effects
2. Line 13, spell FON out when mentioning it the first time.
3. Line 14, spell MDA out when mentioning it the first time.
4. Line 128, isn't CFU a term only for counting bacteria? Could you define CFU and CPU?
5. Is Figure 4 (C) NS, CK or MS? Could the authors provide the picture of NS vs CK vs MS?

Reviewer 3 ·

Basic reporting

I would like to see improved grammar, sentence composition, word choice. Avoid words like "obviously" or "clearly". See attached document with comments.

Experimental design

No comment

Validity of the findings

No comment

Additional comments

See attached document for additional comments on writing. Perhaps consult a writing mentor or tutor to polish the document.

Annotated reviews are not available for download in order to protect the identity of reviewers who chose to remain anonymous.

---

## Round 0.2 · accepted · Accept

Since the previous reviewer never accepted the review. After my careful review of the authors' revised manuscript and the response comments, I believe that the current manuscript is suitable for publication in PeerJ.

"The last sentence of the abstract needs to be revised. It has not been shown which, if any, of the factors mentioned are causal for the increased Fusarium resistance. (So change to something like " Our results indicated that wheat and watermelon intercropping enhanced watermelon growth and decreased the incidence of Fusarium wilt in watermelon. These effects could be due to intercropping inducing physiological changes, regulating soil enzyme activities, and/or modulating soil microbial communities.")

Minor editing for grammar is needed. For example, line 11 "the Fusarium" --> "Fusarium".

Line 12 "alleviate" --> "alleviates".

Generally the writing and grammar is fine, so I don't think professional editing is required"